



# Submarine Landslide Source Modeling using the 3D Slope Stability Analysis Method for the 2018 Palu-Sulawesi Tsunami

Chatuphorn Somphong[1], Anawat Suppasri[1], Kwanchai Pakoksung[1], Tsuyoshi Nagasawa[2], Yuya Narita[2], Ryunosuke Tawatari[2], Shohei Iwai[2], Yukio Mabuchi[2], Saneiki Fujita[1], Shuji Moriguchi[1], Kenjiro Terada[1], Cipta Athanasius[3] and Fumihiko Imamura[1]

1 The International Research Institute for Disaster Science, Tohoku University, Miyagi, 9800845, Japan.

2 Pacific Consultants Company Limited, Miyagi, 9800811, Japan

3 Center for Volcanology and Geological Hazard Mitigation, Geological Agency of Indonesia, Bundung, Indonesia

*Correspondence to*: Chatuphorn Somphong (somphong@irides.tohoku.ac.jp)

**Abstract.** Studies have indicated that submarine landslides played an important role in the 2018 Sulawesi tsunami event, damaging the coast of Palu Bay in addition to the earthquake source. Most of these studies relied on visible landslides to reproduce tsunamis but could still not fully explain the observational data. Recently, several numerical models included hypothesized submarine landslides that were taken into account to obtain a better explanation of the event. In this study, for the first time, submarine landslides were simulated by applying a numerical model based on Hovland's 3D slope stability analysis for cohesion-frictional soils. To specify landslide volume and location, the model assumed an elliptical slip surface on a vertical slope of 27 m of mesh-divided terrain and evaluated the minimum safety factor in each mesh area based on the surveyed soil property data extracted from the literature. The landslide output was then substituted into a two-layer numerical model based on a shallow-water equation to simulate tsunami propagation. The results were combined with the other tsunami sources, i.e., earthquakes and observed coastal collapses, and validated with various postevent field observational data, including tsunami runup heights and flow depths around the bay, the inundation area around Palu city, waveforms recorded by the Pantoloan tide gauge, and video-inferred waveforms. The model generated several submarine landslides, with lengths of 0.2-2.0 km throughout Palu Bay. The results confirmed the existence of submarine landslide sources in the southern part of the bay and showed agreement with the observed tsunami data, including runups and flow depths. Furthermore, the simulated landslides also reproduced the video-inferred waveforms in 3 out of 6 locations. Although these calculated submarine landslides still cannot fully explain some of the observed tsunami data, they emphasize the possible submarine landslide locations in southern Palu Bay that should be studied and surveyed in the future.

## 1 Introduction

Indonesia has recently experienced hardship because of earthquakes and tsunamis. Approximately 2.5 million people have been exposed to a tsunami with a 500 year return period, and a recent study suggested that Indonesia ranks third in the world for the proportion of the population exposed to tsunamis (Løvholt et al., 2014). A total of 9 tsunamis have been observed since 2000, 8 of which were associated with a significant earthquake, including the deadly Indian Ocean tsunami in 2004, according to the National Geophysical Data Center/World Data Service (NGDC/WDS, 2021). These types of natural disasters have caused many economic losses and fatalities in Indonesia. The country experienced limitations in reconstruction after the 2004 tsunami (Suppasri et al., 2015). Horspool et al. (2014) indicated that the annual probability of experiencing a tsunami similar to the aforementioned 3.0 m tsunami is ~0.1-10% across the entire Indonesian coast.



The $M_w$ 7.5 earthquake that occurred on September 28, 2018, near Central Sulawesi, Indonesia (Lat.: 0.256°S, Long.: 119.846°E), was caused by strike-slip faulting at shallow depths within the interior of the Molucca Sea microplate, which is part of the broader Sunda tectonic plate (USGS, 2018). The ruptured fault is a segment of a major active strike-slip fault zone named the Palu-Koro Fault System (Liu et al., 2020). A major surface rupture with a 4 m maximum left-lateral offset was identified

during a field survey in the NW Palu Valley (Patrai et al., 2020). In addition, a 5.8 m surface rupture in Pawunu village occurred approximately 15 km south of Palu city (Banda Geologi, 2018). A unanticipated tsunami was triggered by this earthquake and caused damage around Palu Bay. The cumulative impacts from this tsunami led to approximately 2,100 fatalities and more than 68,000 damaged houses (Pang, 2018; Hadi and Kurniawati, 2018). Several post-earthquake-tsunami field surveys reported damage along the Palu Bay coast and indicated runup heights above 6.0 m (Mikami et al., 2019), specifically ~9.1 m runup

heights and 8.7 m inundation depths at Benen village (Omira et al., 2019) and ~10.7 m runup heights in Tondo (Widiyanto et al., 2019). Outside of the bay, there is little evidence of tsunami flooding-related disruption, implying that most tsunamigenic sources were located within the bay (Liu et al., 2020)

## 1.1 Review of the attempts to model tsunami sources

Many attempts have been made to reproduce Palu tsunami events through modeling; however, these studies produced different

outputs based on various tsunami sources. The finite fault model proposed by the USGS created only small vertical ground displacements that were not large enough to cause significant tsunamis within the bay (Pakoksung et al., 2019). Augmentations to the sources of the tsunami have been made. Due to the difficulty in reproducing the observed tsunami runup heights and the Pantoloan tide gauge record, landslide sources have been proposed as an additional source. At the Pantoloan tide gauge station in Palu Bay, time series of a tsunami waveform was recorded. Several videos of the tsunami's arrival were also captured. Carvajal

et al. (2019) used these videos to calculate the tsunami's arrival times and infer the waveforms at each site and emphasized that coastal/submarine landslides were main contributing sources of this Palu tsunami.

Simulations of earthquake-generated tsunamis and submarine landslide-generated tsunamis involve different approaches. The generation of a tsunami induced by submarine landslides is a complicated process. Coupled dynamic models for earthquake-generated tsunamis have been well established and are commonly used, while coupled dynamic models for landslide-generated

tsunamis are still limited (Heinrich et al., 2001; Pakoksung et al., 2019). Various modeling efforts have been carried out to simulate tsunami propagation using the COMCOT model (Heidarzadeh et al. 2019; Carvajal et al., 2019; Gusman et al. 2019; Liu et al., 2020; Sepúlveda et al., 2020). In more recent works, several studies have been performed to simulate landslide-induced tsunamis using other uncoupled methods, in which the landslide mass and tsunami propagation are separately calculated (Nakata et al., 2020, Paris et al., 2020; Ulrich et al. 2019; Schambach et al., 2021).

Some previous works have successfully acquired agreeable results by comparing the surveyed runup heights to the inundation depths, the recorded water levels at the Pantoloan tide gauge, or video-inferred waveforms. Many previous studies have used numerical models to reproduce a waveform at the Pantoloan tide gauge and have obtained good agreement with recorded data (e.g., Heidarzadeh et al. 2019; Carvajal, M. et al., 2019; Takagi et al., 2019; Jamelot et al., 2019). Additionally, several studies have attempted to model the tsunami runup heights and inundation depths, have compared them to the surveyed data, and have

achieved accepTable comparisons (e.g., Pakoksung et al. 2019; Ulrich et al. 2019). Moreover, some studies have focused on inundation in Palu city. For example, Gusman et al. (2019) performed a simulation using vertical displacement and landslide sources to determine the inundation depths and area and compared the results with high-resolution field measurements by Paulik et al. (2019). More updated studies have attempted to reproduce video footage-based waveforms, as proposed by Carvajal, M. et al. (2019). In some works, hypothesized landslide sources are required to explain video-inferred waveforms (e.g., Nakata et al.,





2020; Schambach et al., 2021). Moreover, combining seismic fault slip with observed landslide sources can also explain the wave characteristics based on video footage. For instance, Sepúlveda et al., 2020 combined landslide sources with coseismic ground displacement measured by Interferometric Synthetic Aperture Radar (InSAR) with different fault geometry configurations and archived the simulated tsunamis that were consistent with all of the tsunami data, including surveyed runup heights and video-inferred waveforms. These studies conducted simulations with different numerical models that used different

sources of tsunamis or sources of landslide data to explain certain types of observational data. Table 1 summarizes the past literature with respect to numerical models, tsunami sources, landslide data sources and verification data.

### 1.2 Review of submarine landslides in Palu Bay

Several studies have observed visible coastal landslides or submarine landslides through various methods. Small submarine landslide sources were identified from eyewitness accounts and field surveys (Arikawa et al. 2018). Sassa and Takagawa (2018)

suggested that liquefaction induced coastal land collapse, which resulted in liquefied sediment flows and eventually led to a tsunami. Carvajal et al. (2019) used video footage and satellite images to identify visible coastal land collapses, and these landslide sources were used by later studies, including Gusman et al. (2019) and Tegaki et al. (2019). Liu et al. (2020) studied coastal landslides by comparing pre-earthquake bathymetry to their surveyed postearthquake bathymetry. These observed landslide locations are mapped and summarized in Fig. 1. Since actual submarine landslides are difficult to detect, hypothesized

landslides have been proposed as an alternative in more recent works. Pakoksung et al. (2019) introduced four mega-sized landslides with volumes of ~10-38 million $m^3$: three in the upper outside of Palu Bay and one in the lower part. Similarly, Nakata et al.'s (2020) study strongly emphasized a potential submarine landslide in the nearshore area near Palu city (Fig. 1).

### 1.3 Research objectives

Earthquake-induced tsunami modeling in previous studies cannot fully explain observational data. A better model for

reproducing these tsunami impacts may be to reproduce tsunami events. Many past studies rely on hypothesized submarine landslides (Takagi et al., 2019; Pakoksung et al., 2019; Nakata et al., 2020) and suggest various potential submarine landslides within Palu Bay. This raises the question of whether exact landslides could actually occur in these areas and requires further investigation regarding these undiscovered submarine landslides. Therefore, more studies should focus on modeling submarine landslide sources.

To the best of our knowledge, no existing studies have considered modeling submarine landslide sources based on slope stability analysis and the available observed soil data. The objectives of this study are to

1)  present probable submarine landslide sources using a sophisticated landslide model based on 3D slope stability analysis (which has never been performed according to the existing literature) that reflects the existing observational soil data, and then check whether the simulated submarine landslides match the observations or are located within potential areas

suggested by past studies.

2)  reproduce all the field observations of tsunami records, including the runup height, flow depth (or inundation depth) around Palu Bay, Pantoloan tide gauge waveform, video-inferred waveforms, and inundation depths and area in Palu city, with the developed landslide model.


## 2 Methodology

This section is composed of three parts. First, the landslide model is introduced; second, the numerical model used for tsunami propagation is introduced. Last, the data used for model settings and observational data used for model verifications are presented.

Figure 2 presents the workflow of this study regarding modeling procedures. Generally, there are 2 models: landslide and tsunami models. The former model considers the significant measured soil parameters as an input, including unit weight,

cohesion, and internal friction angle. The modeling requires the initial water level and groundwater conditions according to the bathymetry. Similar to Nakata et al. (2020) and Paris et al. (2020), the landslide model assumes the initial shape of the landslide mass to be an ellipsoid. The geometry of landslides includes the maxima of the major and minor radii of the ellipsoid and the soil layer thickness. The seismic force is considered an additional sliding force in this study, which is discussed in a later section. Once the parameters are set, the model calculates the landslide mass (or landslide thickness) and the location of the landslide

based on the minimum safety factor; these values are input into the tsunami model. The second part of the modeling procedure simulates tsunami propagation. This study adopts the same methodology as Pakoksung et al. (2019), details of which are explained in a later section. The reproductions of tsunami runup height, inundation depth, Pantoloan tide gauge waveform, video-inferred waveforms at 6 locations, and inundation area in Palu city are determined to validate the observational data. The Comparison Index (Aida, 1978) is used to quantitatively compare the measured and calculated runup heights and inundation

depths. The input soil parameters and landslide geometry are repeatedly calibrated and adjusted until the Comparison Index reaches agreeable values.

### 2.1 Landslide model

### 2.1.1 Model description

This study adopts a model based on Hovland's (1977) method as the landslide model. In Hovland's method, the collapsed soil

mass cut off by the slip surface is decomposed into square or rectangular soil columns. Hovland's method defines the three-dimensional factor of safety as the ratio of the total available resistance along the failure surface to the total mobilized stress along it (Ahmed et al., 2012). The safety factor formula for Hovland's method is shown in Eq.1:

$$F = \frac{\sum_x \sum_y [cA + \gamma \ \Delta x \Delta y \cos(DIP) \tan \emptyset]}{\sum_x \sum_y \gamma z \Delta x \Delta y \sin \alpha_{yz}} \tag{1}$$

where $F$ is the safety factor, $c$ is the adhesive strength, $\phi$ is the internal friction angle, $A$ is the area of the slip surface, $\Delta x$ and $\Delta y$

are the soil mass length and width when the column is projected onto the $xy$ plane, $\alpha_{yz}$ is the angle formed by the slip plane and the $yz$ plane, $DIP$ is the inclination direction of the slip surface, $z$ is the depth of the soil mass, and $\gamma$ is the unit weight of the soil mass. From the above, the formula can be simplified as the proportion of the resistance of the entire slipping soil mass, $Q = \Sigma Q_i$, over the sliding force, $S = \Sigma S_i$, as Eq. 2:

$$F = \frac{\sum Q_i}{\sum S_i} \tag{2}$$

For stability over seismic activity, the design horizontal seismic intensity is introduced to consider the influence of the earthquake. The horizontal force due to the earthquake motion is assumed to be a part of the weight of the soil mass in the horizontal direction, as shown in Fig. 3a. The design horizontal seismic intensity is given as a vector $k$ in the same horizontal direction as the gradient direction of the calculated element for each soil column. At this time, the resistance force ($Q_i$) and sliding force ($S_i$) of each soil mass are described as follows.

$Q_i = [cA_i + \{n_i \cdot (e_g + k)(t_i \cdot u_i)(W - W_i)\} \cdot \tan \phi] u_i \tag{3}$



$$Si = - [t_i \cdot (e_g + k)(W_i)]t_i \tag{4}$$

where $A_i$ is the area of the sliding surface, $W_i$ is the weight of the earth column, and $n_i$ and $t_i$ are the unit vectors perpendicular to and parallel to the sliding surface, respectively. $u_i$ is the direction in which the slip body slides and is determined at the same time as the slide body. $e_g$ is the unit vector in the direction of gravity (Fig. 3b).

Generally, the input soil parameters that are typically required for numerical modeling are soil unit weight, tensile strength, Young's modulus, Poisson's ratio, cohesion and angle of friction (Kumar et al., 2016). In this study, the landslide model requires 3 essential soil parameters: the saturated condition of the cohesion, the internal friction angle and the unit weight of the soil. In this study, landslide failure was analyzed by considering a safety factor < 1.

### 2.1.2 Measured soil data and landslide model settings

Observations of the submarine soil properties in Palu Bay do not exist in the literature. This research adapts soil data from field observations near the bay. The soil properties are defined by the pressure- and shear-wave velocity ($V_p$ and $V_s$) profiles inferred from horizontal-to-vertical spectral ratio (HVSR) microtremor inversion over the soil depth, which was proposed by Cipta et al. (2020). Subsequently, $V_s$ is translated to the standard penetration test (SPT) count ($N$) by the empirical equation proposed by Kirar et al. (2016) for all soil types: $V_s = 99.5N^{0.345}$. Overall, based on the measured soil properties, the study area is categorized

into 3 main parts: upper, middle and lower Palu Bay, as shown in Fig. 4. By the mechanism in the landslide model, the soil layers are categorized into 2 main layers: the sliding layer(s) and base layer. Land failure is allowed on only the sliding layers (as shown by the stratum in Fig. 5). This research considers only 2 stratums based on the observed soil properties in Fig. 4 and applies those from observation sites 504, 402 and 1001 for strata in the Upper Bay, Middle Bay and Lower Bay, respectively. In addition, the soil properties are assumed to be saturated soil conditions, which significantly affect the result more than dry soil

conditions in the landslide model.

After determining the SPT $N$ value, the saturated soil cohesion and internal friction angle are calculated from the empirical equations proposed by Kumar et al. (2016). The cohesion ($C$) can be computed from $C = -2.2049 + 6.484N$, for $N$ ranges from 2-30 (for N < 2, we assume a random low $C$ value). Moreover, the angle of internal friction ($\varphi$) is calculated from $\varphi = 7N$ for $N < 4$. $\varphi = 27.12 + 0.2857N$ when $N$ is in the range of 4-50. The values of these soil properties are tested to find good agreement with

the observed tsunami runup height and inundation depth and can be slightly different from the actual computed values from the above equations. Groundwater is assumed to exist, and the groundwater level is equal to the seabed elevation for all areas. Based on the observed water level at the Pantoloan tidal gauge, the initial water level is set to 2.3 m above mean sea level (MSL), representing high tide conditions (Pakoksung et al., 2019).

Because of the unknown exact value of soil properties that can be estimated from only the stated empirical equations, this study

considers a variation in soil parameters according to the available soil property data (Fig. 4). For example, the measured soil unit weight varies from ~11.7-12.4 kg/m$^3$ for stratum no. 1 (the first ~3-8 m of depth from the ground surface) in Lower Palu Bay and ~13.2-15.0 kg/m$^3$ for stratum no. 2. Regarding the landslide geometry, this study assumes an elliptical sliding surface. The geometry is composed of a major radius (the longer radius of the ellipse) and a minor radius. The major radius is always set equally to 2 times the minor radius. This study varies the major radius from 0.5 km to 1 km. We perform ~70 configurations

based on these properties, and the final parameter settings for the landslide model are summarized in Table 2.

### 2.1.3 Seismic force

To evaluate the impact of the earthquake, the design horizontal seismic intensity was proposed by the Eq. 5 (Noda et al., 1975):




$$|k| = \frac{1}{3}\left(\frac{a}{g}\right)^{1/3}$$ (5)

where $k$ is the design horizontal seismic intensity, $a$ is the maximum value of the ground surface acceleration in cm/s², and g is

the gravitational acceleration, which is equal to 980 cm/s². At the Palu site, the ground surface acceleration is approximately 75-97% of the gravitational acceleration (https://earthquake.usgs.gov/earthquakes/eventpage/us1000h3p4/map), which means $k$ = 0.28-0.33. In this study, a $k$ of 0.30 is applied equally to all Palu Bay areas.

### 2.2 Tsunami model

This study adopted the same tsunami model as used in Pakoksung et al.'s (2019) study. A two-layer computational model,

TUNAMI-N2, was developed to solve nonlinear shallow-water equations with two interfacing layers and kinematic and dynamic boundary conditions at the seafloor, interface, and water surface (Imamura and Imteaz, 1995). The mathematical model used in the TUNAMI-N2 code is made up of a stratified medium of two layers. The first layer is composed of a homogeneous inviscid fluid with a constant density $\rho_1$ that represents seawater, while the second layer is composed of a fluidized granular substance with a density $\rho_s$ and a porosity $\mu$ that represents air. The mean density of the fluidized debris is assumed to be constant and

equal to $\rho_2 = (1 - \mu)\rho_s + \mu\rho_1$ (Macías et al., 2015). A more detailed explanation is given in Pakoksung et al.'s (2019) study.

### 2.3 Topographic data

Regarding the depth of the considered areas, this analysis used bathymetric data supported by Badan Informasi Geospasial (BIG), Indonesia. This database was developed prior to the Palu tsunami of 2018. BIG provided bathymetric and topographic data for the entire Palu Bay region as well as the adjacent continental areas. The domain was set to a 27 m bathymetric resolution

in the sea, yielding a domain of 1155 × 810, which covered the entire Palu Bay area (Fig. 1), and a 1 m resolution for the nearshore Palu city with the size of 4.051 × 1.390 km², resulting in a 4051 × 1390 grid. A constant-grid tsunami simulation was solved at each time step of 0.01 s. It should be noted that the terrain data used in this study are adjusted by adding the constant of 2.3 m due to the lack of inundation area at high tide conditions, as recommended by Pakoksung et al. (2019).

### 2.4 Comparison index

To quantitatively compare the measured and estimated runup heights and inundation depths, Aida's (1978) correlation values; the geometric mean; and $K$ and its variance, $\kappa$, are used and described by Eq. 6 and Eq. 7 respectively:

$$\log K = \frac{1}{n}\sum_{i=1}^{n} \log k_i$$ (6)

$$\log \kappa = \left[\frac{1}{n}\sum_{i=1}^{n}(\log k_i)^2 - (\log K)^2\right]^{0.5}$$ (7)

where, $k_i = R_i/H_i$. $R_i$ is the field measurement's runup height at point $i$. $H_i$ is the calculated runup height at point $i$ from the

simulation, and $n$ is the total amount of data. A $K$ value close to 1 and small $\kappa$ indicates good agreement between the observations and simulations.

### 3 Simulation results

### 3.1 Submarine landslides

This study configured ~70 cases with variations in soil properties, angle of inclination, and earthquake intensity for the best

estimation. A total of 23 submarine landslide events were simulated and are shown in Fig. 6, excluding the proposed





coastal/submarine landslides collected from past studies. The maximum landslide thickness varied from ~5-20 m. Due to the stronger soil properties, fewer submarine landslides appeared in the Upper Bay, more land failure occurred in the Middle Bay, and the Lower Bay zone was the most vulnerable to the earthquake due to weaker input soil properties. The largest submarine landslide occurred around the nearshore area of Palu city, with a size of ~0.04 km$^3$, which is much smaller than the submarine

landslides proposed by Pakoksung et al. (2019) and Nakata et al. (2020).

However, most submarine landslides occurred in the potential areas proposed by past research. Haidarzadeh et al. (2019) proposed possible major submarine landslides along underwater slopes by using backward tsunami ray tracing, in which they placed a point tsunami source at the location of the Pantoloan tide gauge and propagated the tsunami model for 5 minutes (Fig. 6, dashed rectangle). Based on slope stability analysis, this study provided a maximum of 7 submarine landslides that could occur

at that location. The largest computed submarine landslide was located around the area suggested by Nakata et al. (2020), as shown in Fig. 6. The lower blue dashed ellipses hint at a large submarine landslide with a volume of 0.07 km$^3$. Moreover, this study found 3 submarine mass failures with a total size of ~0.05 km$^3$ in the mentioned area. By following Nakata et al. (2020), Schambach et al. (2021) modeled submarine mass failure with a size of ~0.026 km$^3$ in a similar area and found good agreement with the nearby observed runups (Fig. 6, yellow ellipse). According to these resulting landslides, this study strongly emphasizes

the potential submarine landslides in Lower Palu Bay. In addition, the computed submarine landslides in the mentioned lower dashed ellipse area follow the possible erosion zone after the earthquake occurred when compared to the surveyed postearthquake bathymetry based on Liu et al. (2020).

Unlike the study by Pakoksung et al. (2019), who suggested hypothesized submarine landslides in a northern area outside of Palu Bay, this study found only 2 small mass failures around these locations. This study did not generate any submarine landslide in

the upper blue dashed ellipse zone around the Pantoloan tide gauge (Fig. 6), as proposed by Nakata et al. (2020). Moreover, the landslide model failed to simulate landslides in western Palu, as suggested by Carvajal et al. (2019) (zones *I, J, K, L,* and *M*) based on slope stability analysis and the available soil properties. However, Sepúlved et al. (2020) insisted on these contributions as the main sources. Therefore, this study needs to combine these submarine landslides with the present simulations to reproduce tsunami events.

### 3.2 Simulated tsunamis

#### 3.2.1 Runup heights and inundation depths

Figure 7 shows the simulated tsunami runup height and the comparison between simulated and observed values. This study compares the calculated runups with the observational runups from government reports and published research, including (1) the Geological Agency of Indonesia (Badan Geologi, 2018)'s report, (2) The agency of Meteorology, Climatology and Geo-physics

(BMKG) of Indonesia, (3) Omira et al. (2019), and (4) Widiyanto et al. (2019). These studies' surveys cover all of the study area of this paper around Palu Bay. Overall, the simulated tsunami runup heights vary from ~3.5 to 13.5 m on the west coast and ~4.1 to 11.0 m on the east coast. Large wave amplitudes are found at the shoreline of the western coast in Middle Palu Bay and the eastern coast in Lower Palu Bay, which indicate slight overestimation in these areas. The comparison index, including the geometric mean $K$ and geometric standard deviation $\kappa$ is featured in Fig. 7b. According to the figure, overall, the simulation is

overestimated as $K = 1.18$, which is > 1.0, and $\kappa = 1.44$ from 98 observations. A comparison with Widiyanto et al. (2019) gives the best results when $K = 1.17$ and $\kappa = 1.39$ from 27 observation points. The scatter plot in Fig. 7b indicates that most of the computed runups are in the range of $\pm 2.1$ m error. It should be noted that the measured runups from past research were adjusted to our terrain dataset; that is, 2.3 m was added (Pakoksung et al., 2019).



The tsunami inundation based on simulated submarine landslides is also evaluated using the evidence of inundation surveyed by
past studies, including Badan Geologi's report, Omira et al. (2019), and Widiyanto et al. (2019). The tsunami flow depth
apparently follows a similar trend with the runup heights but with more fluctuations. Generally, there are overestimated trends on
both sides of the bay (Fig. 8a), and the simulation is slightly overestimated at $K = 1.13$, which is $> 1.0$. and $\kappa = 2.15$ from 62
observations according to Fig. 8b. The inundation depth results show large variances when compared to the surveyed
inundations. This study calculates the inundation depths by the simulated runup height subtracted by the topographic terrain with
a 27 m resolution. The coarser resolution may inevitably affect the simulation results. This study also performs a tsunami
inundation simulation on Palu city with a 1 m resolution terrain data, which shows a significantly better result. We discuss this
result in the next section.

### 3.2.2 Inundation area and depths around Palu city

Palu city, where many buildings for residential, commercial, industrial, governmental, short-term stay and other purposes are
located, has been impacted by inundation ~300 m from the coastlines. Paulik et al. (2019) conducted a high-resolution field
survey to measure tsunami inundation heights at 371 building sites and indicated inundation depths ranging from 0.1 to 3.65 m,
with a mean of 1.05 m and a standard deviation of 0.55 m. Moreover, the simulation in this study can slightly overestimate. The
calculated inundation depths range from 0.12 to 3.83 m, with a mean of 1.30 m and a standard deviation of 0.50 m above the
ground. A comparison between the simulated and observed tsunami inundation depths is shown in Fig. 9a. A total of 164
observation spots were selected in this study due to the limited availability of 1 m resolution terrain data. The comparison index
indicates that $K = 1.17$ and $\kappa = 1.93$, which means that the depths obtained by the simulations slightly outnumber Paulik et al.'s
(2019) observed inundation depths (Fig. 9b). Although the comparison was made at different scales, the high resolution of the
data resulted in a higher accuracy of the inundation depths.

The boundary of the tsunami inundation area was retrieved from Gusman et al. (2019), who interpreted the boundary from
visible tsunami debris in satellite images. The comparison between the observed inundation area in Palu city is shown in Fig. 9c
and indicates that the calculated inundation area is ~1.188 km², while the selected observation of the inundation area is ~0.747
km². This means that the simulation result is overestimated by ~59%.

### 3.2.3 Pantoloan tide gauge waveform

The tsunami wave amplitude time series at the Pantoloan tidal gauge with detided sea level is depicted in Fig. 10. The first
negative tsunami wave peak of 1.35 m was reached at the tidal gauge within 6 minutes, followed by the first positive peak of
1.93 m 2 minutes later. Several past studies attempted to reproduce the waveform at the Pantoloan tide gauge. Although there
was an argument that the reliability of the 1 minute sampling interval of the tide gauge record did not sufficiently capture the
shorter-period characteristics of tsunamis (Carvajal et al. (2019)), many studies have shown agreeable results by reproducing the
water level observations at the Pantoloan tide gauge. For example, the model of Haidarzadeh et al. (2019), using purely tectonic
sources, performed backward tsunami ray tracing and modeled 5 minute tsunami propagation to indicate 2 potential submarine
landslide areas (Fig. 6) to match the waveform at the location of the Pantoloan tide gauge. Nakata et al. (2020) assumed a
submarine landslide source near the tide gauge (Fig. 6, upper blue dashed ellipse) to explain the waveform characteristics.
Moreover, Pakoksung et al., 2019 suggested that large assumed submarine landslides should be located north outside of the bay
to generate the observed waveform characteristics at the Pantoloan tide gauge. Therefore, with calculated submarine landslides
based on Hovland's slope stability analysis, this study supports the hypothesized area of Haidarzadeh et al. (2019) and shows
that the landslides located in the southern part of Palu Bay affect the water level at the Pantoloan tide gauge. However, the results



do not show the absolute accuracy of the peak tsunami wave. The proposed submarine landslide sources combined with the observed landslides generate a sooner arrival time and larger peaks of tsunami waves than the recorded data. The simulated waveform has an extreme negative peak of -3.0 m at 3 minutes and a positive peak of 2.15 m at 5 minutes after landslide

occurrence (Fig. 10). The peak error percentage is used to indicate the performance of the simulation and can be determined by Eq. 8:

$$\% \ Error_{peak} = \frac{Peak_{simulated} - Peak_{observed}}{Peak_{observed}} \times 100 \tag{8}$$

where $Peak_{simulated}$ represents the peak calculated from the model simulation and $Peak_{observed}$ is the peak of the tsunami wave from the observations. According to the equation, the simulation yields a peak error of 11.40% and a 3 minute shorter arrival time. It

should be noted that this study tried to simulate only the observed submarine landslide sources (as was conducted in Lui et al., 2020), and the results did not show good agreement with the observed waveform.

### 3.2.4 Video-inferred waveforms

Past studies have shown some difficulties in reproducing CCTV-based tsunami characteristics based solely on coseismic sources for every available footage since they were introduced as an observational dataset (Carvajal et al., 2019; Sepúlveda et al., 2020).

The observed tsunami waveforms inferred from video footage have short arrival times, and periods of ~2 minutes resulted in short wavelengths (Carvajal et al., 2019). In general, by visual observation, the results of simulated video-inferred waveforms have even longer arrival times than the waveforms inferred from CCTV footages by judging the first peak of the tsunami wave. Three out of 6 video inferred waveforms had comparable characteristics in southern Palu Bay, including in Talise, the KN Hotel area, and West Palu (Fig. 11), while the simulation results cannot fully explain the observations in northern Palu Bay, Pantoloan

and Wani. The simulation demonstrates that an overestimation of calculated landslide sources resulted in more devastating tsunami waves with higher peaks and longer wave periods in southern Palu Bay. The peak errors are 165%, 22%, -39%, 32%, 36% and -8% for Dupa, the KN Hotel, Pantoloan, Talise, Wani and West Palu, respectively. The time of the peak in Dupa, Pantoloan, Wani and West Palu is approximately 0.45, 1.50, 2.95 and 0.21 minutes before the observations, while the peak time in the KN Hotel and Talise occur shortly after the observation, by -1.30 and -0.10 minutes, respectively. However, there is

uncertainty regarding the timing of the video-inferred time series because the video footage was apparently captured promptly after the tsunami rather than earthquake shaking. The uncertainty may also result in ±15 to ± 30 s of arrival time error in this study (Sepúlveda et al., 2020).

### 4 Discussions

Regarding the model parameters for simulating submarine landslide sources, we configured the model parameters, i.e., statured

soil unit weight, cohesion, internal friction of angle, groundwater condition, mean sea level and earthquake intensity, as close as possible to the existing data. Slope failures can be found only in some steep slope areas inside Palu Bay. The model fails to comprehensively reproduce the observed landslides, which are located near Palu Bay, for both location and mass size. Based on Hovland's slope stability analysis and measured soil properties, there is no slope failure in the mentioned nearshore or coastal areas. Despite modeling failure, this study can simulate submarine landslides in the potential areas proposed by Haidarzadeh et

al. (2019), Nakata et al. (2020), and Schambach et al. (2021), which can imply that based on the supported theory, undiscovered submarine landslides in those areas are possible, especially a large mass failure with a size of at least 0.05 km³ in southern Palu Bay (Fig. 6, blue dashed ellipse). However, this study overproduced submarine landslide sources in southern Palu Bay according to tsunami simulation results. The runup heights, inundation depths, waveforms at the Pantoloan tide gauge, video footage-based





waveforms, inundation area and depths in Palu city all generally show overestimated trends. In particular, a comparison with
observations shows obvious overestimation of the simulation results of the southern Palu areas. These results imply that the real
soil properties have a strong spatial variation. The soil properties in the simulation of southern Palu Bay, which was sampled
near the Palu River, are probably weaker than the realistic conditions, resulting in many larger submarine mass failures than the
observed failures. In addition, this study cannot reproduce the landslide sources around nearshore Palu Bay, where past studies
strongly indicate that they mainly contribute to the Palu tsunami, i.e., zones *I, J, K, L,* and *M* (Carvajal et al., 2019; Sepúlveda et
al., 2020), and zones *B* and *C* (Liu et al., 2020; Schambach et al., 2021) because these areas probably have stronger soil
properties than those in the simulation. Notably, a landslide source in the nearshore area around the Pantoloan tide gauge, as
proposed by Nakata et al. (2020) (Fig. 6, upper blue dashed ellipse), which still cannot be explained by this study, also needs to
be addressed.

This study applied a constant earthquake intensity to the entire domain; however, in reality, earthquake intensity varies spatially.
The peak intensity was located in southeastern Palu Bay according to the USGS. It should be noted that we tried to vary the
earthquake intensity between 3 study zones, with a maximum intensity of 76% for Upper Palu and 80% intensity for Middle and
Lower Palu Bay, but the results did not show a significant difference from those when a constant intensity was applied.

There is controversy regarding the Hovland method. Hovland's slope stability ignores the resistant force acting on the surface
and the friction between soil columns and considers only the force acting on the slip surface. This leads to a smaller safety factor
when compared to the other methods (e.g., Janbu) (Ahmed et al., 2014); in other words, Hovland's method tends to give an
overestimated output. However, this method is simple and suiTable for calculation in a large area or a high-resolution terrain.
There are alternative methods to tackle the problem of overestimating submarine landslides, such as the Bishop and Janbu
methods. Hungr et al. (1989) stated that the simplified Bishop method offers a simple and efficient alternative that is applicable
to a wide range of practical problems.

Further field observations of submarine landslide sources are required to confirm this study's results, especially for submarine
landslides in the deep Lower Palu Bay, as suggested by past studies and emphasized by this study. Field observations are also
needed to confirm the soil property estimations used in the study area.

Further research should focus on the variation in the time of landslide initiation. This study assumed that all slope failures occur
at the same time and start immediately after the earthquake, which may not reflect reality, as there is potentially a difference in
the time at which a landslide starts to move. Adding coseismic sources or applying the results from the fault-slip model may help
improve this study; Sepúlveda et al., 2020 showed that the combination of earthquake-triggered landslides and the inverted
coseismic ground displacements measured with InSAR can reproduce the Palu tsunami with good agreement.

## 5 Conclusions

Rather than employing a trial-and-error approach to the generation of hypothesized landslide sources, this study applied a
landslide model based on slope stability analysis for the first time to provide a better understanding of the potential submarine
landslide-induced tsunami phenomenon. The simulated tsunami that was triggered by modeled submarine landslide sources
based on Hovland's 3D slope stability analysis with observational soil property data and configured landslide geometry shows
that the submarine landslide in southern Palu Bay is consistent with the past study; however, the magnitudes of simulated
submarine landslides are overestimated, resulting in an overestimation of tsunami parameters, including runup height and
inundation depths, around Palu Bay and Palu city. This implies that the measured soil parameters inland of Palu Bay used in this
study cannot fully explain the observed landslides. It is difficult to validate the simulation results according to all the available



observational data, including tsunami runup and inundation depth around Palu Bay and Palu city, water level at Pantoloan tide gauge, and video footage-based waveforms. The simulation results still do not fully explain the observational data, especially the flow depth across Palu Bay. Field observations of submarine landslides, especially in southern Palu, are needed to confirm the

results of this study. In future studies, more accurate landslide simulations should be used by applying a more realistic configuration of the landslide thickness, soil parameters and landslide geometry or dividing the study zones into zones with finer resolution. Additionally, the various landslide occurrence times should also be investigated, as strongly recommended by previous literature.

**Data availability**

The data used in this study could be requested from the corresponding authors. The author would like to support the interested researchers for further research.

**Author contributions**

CS and AS initiated the original idea and conceptualized research. CS and KP performed the modelling with input from CA. SF, SM and KT supported the original code of landslide model. KP, FI supported the tsunami propagation code. The simulation

results were discussed among all the co-authors. CS wrote the draft with the contribution from AS, TN, YN, RT, SI, YM.

**Competing interests**

The authors declare that they have no conflict of interest.

**Acknowledgments**

The authors would like to thank the Center for Volcanology and Geological Hazard Mitigation, Geological Agency of Indonesia

for providing the observational data and tsunami flow depth data used to validate the tsunami models of the 2018 Palu tsunami, and the Coastal Disaster Mitigation Division, Ministry of Marine Affairs and Fisheries, Jakarta, Indonesia for issuing tide gauge records at Pantoloan. A special thanks to Japan International Cooperation Agency (JICA) for the support of high resolution of digital elevation model.

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



**Table 1 Summary of the main characteristics of the 2018 Palu event and tsunami reported in previous literature**

| Study | Numerical Model(s) | Source of Tsunami | | Source of Landslide Data | | Verification Data | | | |
| --- | --- | --- | --- | --- | --- | --- | --- | --- | --- |
| | | Landslide | Earthquake | Observations | Assumed/ Hypothesized Landslides | Tide Gauge | Inundation area | Run-up Height/ Inundation depth | Video Footage |
| Heidarzadeh et al. 2019 | COMCOT | | ○ | | | ○ | | | |
| Carvajal, M. et al., 2019 | COMCOT | | ○ | ○ | | ○ | | ○ | |
| Takagi et al., 2019 | Delft-3D | | ○ | | ○ | ○ | | ○ | |
| Pakoksung et al. 2019 | TUNAMI-N2 | ○ | | ○ | ○ | ○ | ○ | ○ | |
| Jamelot et al.,2019 | NLSWE (Heinrich *et al.* (1998) and Hebert *et al.* (2001)) | | ○ | | | ○ | ○ | ○ | |
| Ulrich et al. 2019 | Coupled EQ + tsunami model, Seisol + StormFlash2D | | ○ | | | ○ | | ○ | |
| Gusman et al. 2019 | COMCOT | ○ | ○ | ○ | | ○ | ○ | ○ | |
| Goda et al., 2019 | NLSWE (Goda *et al.*, 1997) | | | | | ○ | ○ | | |
| Nakata et al. 2020 | JAGURS | ○ | ○ | | ○ | ○ | | ○ | ○ |
| Liu et al., 2020 | COMCOT | ○ | | ○ | | ○ | | | ○ |
| Sepúlveda et al., 2020 | COMCOT | ○ | ○ | ○ | | ○ | | ○ | ○ |
| Schambach et al., 2021 | NHWAVE/FUNWAVE | ○ | ○ | ○ | ○ | ○ | | ○ | ○ |
| Present Study | Numerical model based on Hovland's (1977) + TUNAMI-N2 | ○ | | ○ | ○ | ○ | ○ | ○ | ○ |





**Table 2 Parameter settings for the landslide model**

| Soil Properties | Upper Bay | Lower Bay | Middle Bay |
|---|---|---|---|
| **Soil Parameter (Stratum no. 1)** | | | |
| Stratum depth (m) | 3.0 | 6.0 | 3.0 |
| Saturated Condition | | | |
|   -   Cohesion (kPa) | 12.0 | 7.5 | 5.0 |
|   -   Internal Friction Angle (degree) | 15.0 | 10.0 | 7.0 |
|   -   Unit Weight (kN/m$^3$) | 11.5 | 12.4 | 11.7 |
| **Soil Parameter (Stratum no. 2)** | | | |
| Stratum depth (m) | 7.0 | 14.0 | 12.0 |
| Saturated Condition | | | |
|   -   Cohesion (kPa) | 17.5 | 23.7 | 20.0 |
|   -   Internal Friction Angle (degree) | 24.0 | 28.3 | 24.0 |
|   -   Unit Weight (kN/m$^3$) | 14.3 | 13.2 | 13.0 |
| **Underground** | | | |
| Groundwater Table Offset (m) | 0.0 | 0.0 | 0.0 |
| Mean Sea Level (+msl) | 2.3 | 2.3 | 2.3 |
| **Base layer** | | | |
| Dry Condition | | | |
|   -   Cohesion (kPa) | 50.0 | 50.0 | 50.0 |
|   -   Internal friction Angle (degree) | 30.0 | 30.0 | 30.0 |
| Saturated Condition | | | |
|   -   Cohesion (kPa) | 20.0 | 20.0 | 20.0 |
|   -   Internal Friction Angle (degree) | 25.0 | 25.0 | 25.0 |
| **Ellipsoid Geometry** | | | |
| Maximum Major Radius | 30 | 30 | 30 |
| Actual Length (m) | 810.0 | 810.0 | 810.0 |
| Minimum Minor Radius | 15 | 15 | 15 |
| Actual Length (m) | 405.0 | 405.0 | 405.0 |
| **Earthquake Intensity** | | | |
| Acceralation (%g) (g = 980 cm/s$^2$) | 70% | 73% | 70% |
| Horizontal seismic intensity, k | 0.30 | 0.30 | 0.30 |




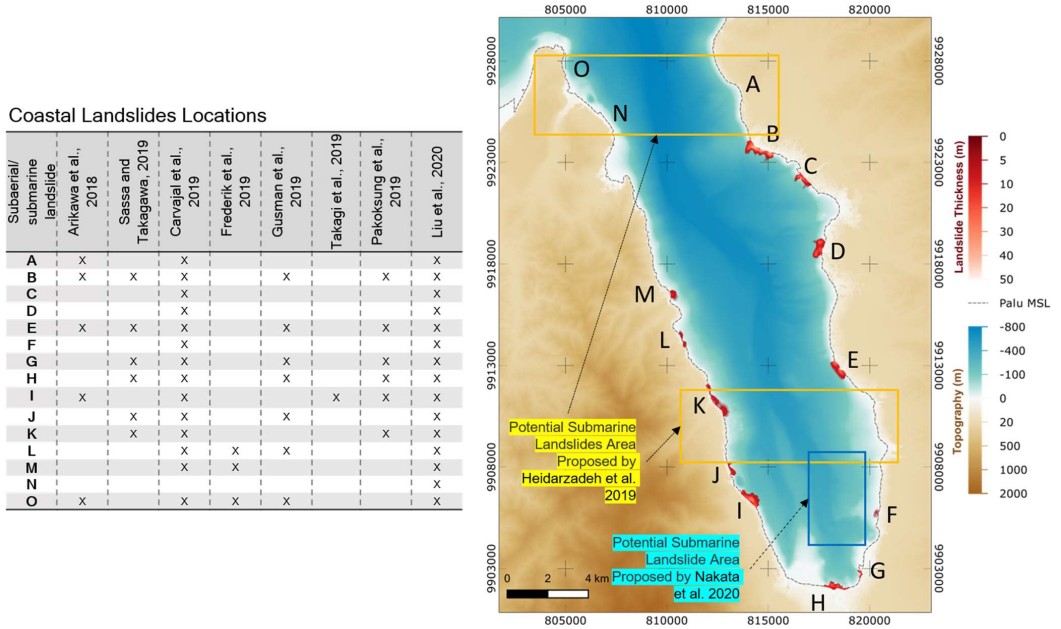

**Figure 1** Literature review of observed landslides: A-O represents the locations of observed coastal landslides, and the Table summarizes landslides used/proposed in each study. The areas defined by yellow rectangles depict the potential submarine landslide areas proposed by Haidarzadeh et al. (2019), and the blue rectangle depicts the potential submarine landslide area proposed by Nakata et al. (2020).

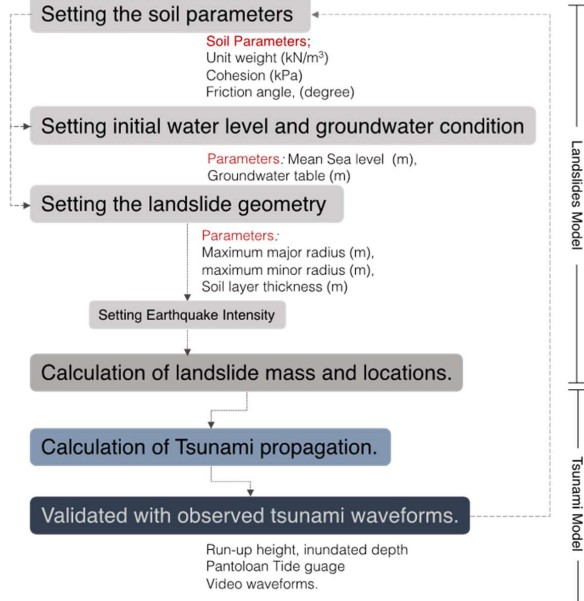

**Figure 2** Diagram of modeling procedures.


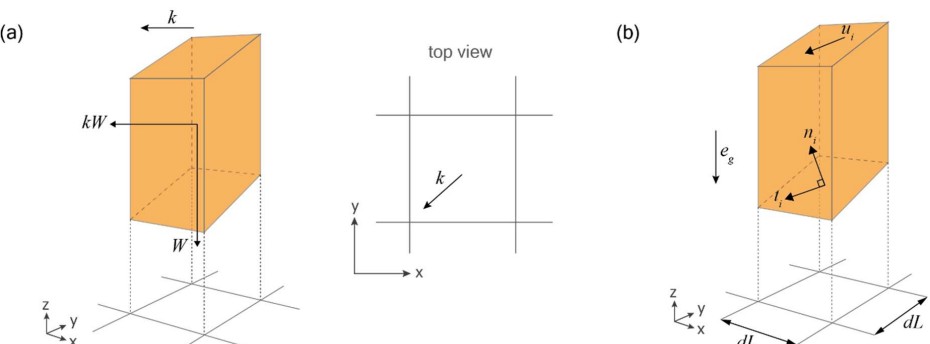

**Figure 3** Diagram of soil mass components; (a) design horizontal seismic intensity and direction of the soil mass. (b) Shape of the soil mass
and directional vectors used for calculation.

| Site no. 504 | | | | | |
|---|---|---|---|---|---|
| Thickness (m) | Depth (m) | $V_p$ (m/s) | $V_s$ (m/s) | Unit Weight (kg/m³) | STP Blow Counts, $N$ (Kirar et al.,2016) |
| 1.24 | 1.24 | 317.85 | 169.59 | 12.42 | 4.69 |
| 1.75 | 2.99 | 569.24 | 257.80 | 13.32 | 15.79 |

| Site No. 402 | | | | | |
|---|---|---|---|---|---|
| Thickness (m) | Depth (m) | $V_p$ (m/s) | $V_s$ (m/s) | Unit Weight (kg/m³) | STP Blow Counts, $N$ (Kirar et al.,2016) |
| 1.06 | 1.06 | 325.57 | 106.13 | 11.30 | 1.21 |
| 1.39 | 2.45 | 383.94 | 196.24 | 11.56 | 7.16 |
| 3.33 | 5.78 | 495.40 | 205.06 | 17.07 | 8.13 |

| Site No. 1001 | | | | | |
|---|---|---|---|---|---|
| Thickness (m) | Depth (m) | $V_p$ (m/s) | $V_s$ (m/s) | Unit Weight (kg/m³) | STP Blow Counts, $N$ (Kirar et al.,2016) |
| 1.55 | 1.55 | 106.29 | 64.58 | 10.89 | 0.29 |
| 7.03 | 8.58 | 272.78 | 153.62 | 11.83 | 3.52 |
| 2.05 | 10.63 | 433.74 | 200.40 | 13.10 | 7.61 |

**Figure 4** Soil condition data for observation site nos. 1001, 402 and 504 and the Paly Bay zones delineated by the soil properties. The Table
includes the shear-wave velocity profile P ($V_p$) and S ($V_s$) parameters inferred from HVSR microtremor inversion, as well as the soil density
over the soil depth and the soil thickness in each layer.


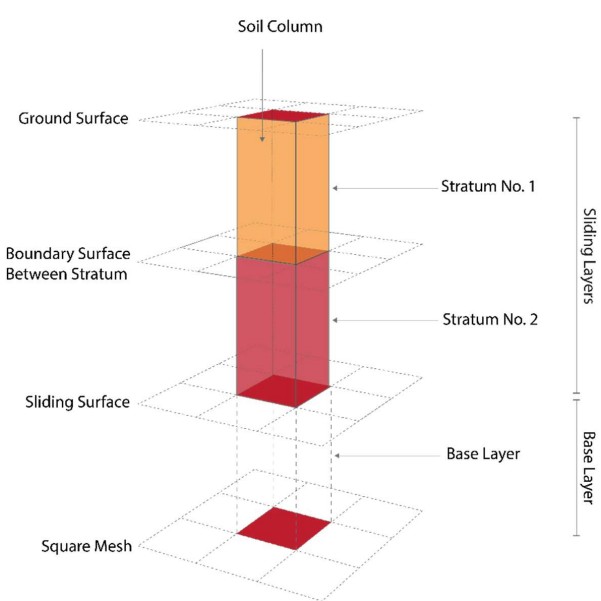


**Figure 5** Diagram of the underground soil column structure in the model.

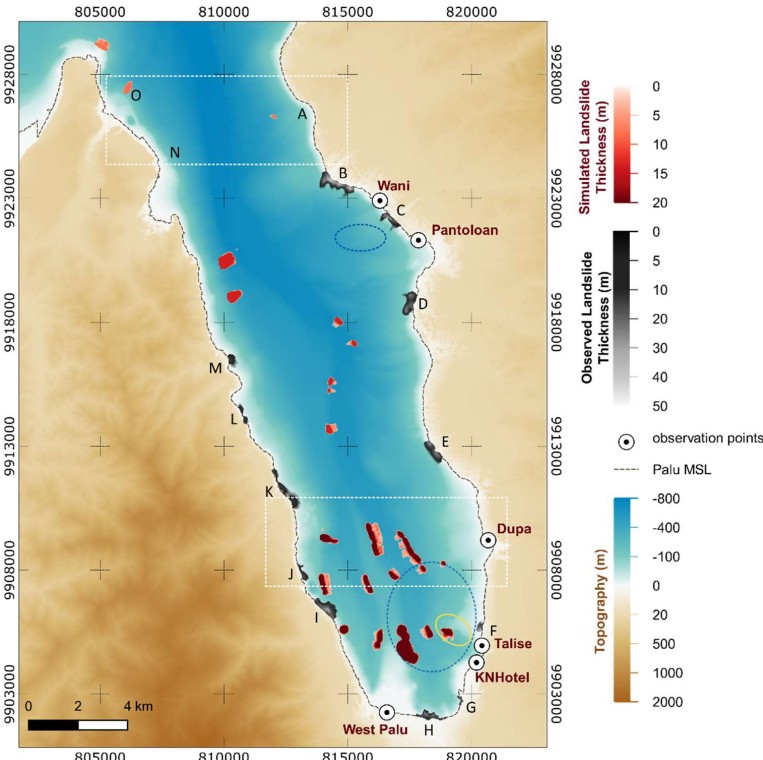


**Figure 6** Simulated submarine landslides calculated based on Hovland's slope stability analysis and surveyed coastal/submarine landslides in previous literature. The white and red color bar indicates the simulated landslide thickness for each grid (27 m resolution), and the black and white color bar indicates the observed landslide thickness. The dashed rectangles represent the potential submarine landslide areas proposed by Haidarzadeh et al. (2019), the dashed ellipse represents that proposed by Nakata et al. (2020), and the yellow ellipse represents that proposed by Schambach et al. (2021). The dots represent the video-inferred waveform observation points. A-O indicate the observed landslides collected from previous studies.

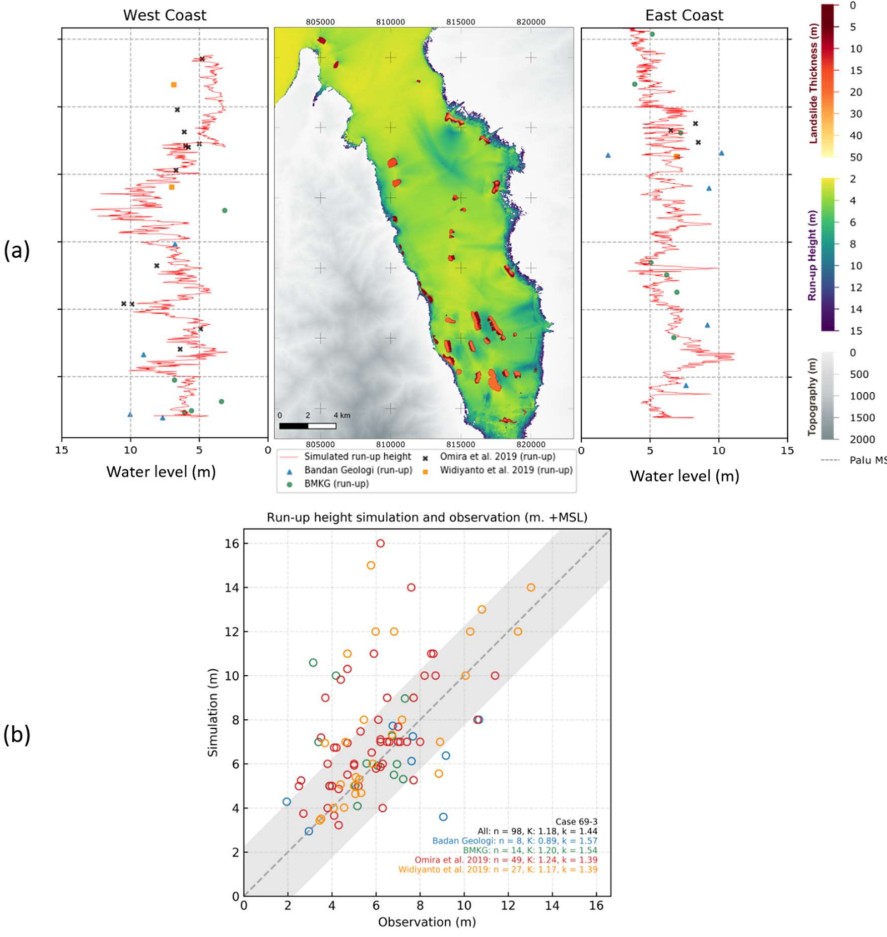


**Figure 7** Simulated tsunami runup heights resulting from the modeled submarine landslides based on Hovland's slope stability analysis and combined with past studies' observed submarine landslides. (a) Calculated maximum tsunami amplitude (center) with the maximum calculated tsunami amplitude (red line) at the shoreline of the western coast (left) and the eastern coast (right), and the measured runup heights from Badan Geologi's report (blue triangles), BMKG (green circles), Omira et al. (2019) (crosses), and Widiyanto et al. (2019) (yellow squares). (b) Comparison between simulated and observed tsunami runup heights by Badan Geologi's report (blue), BMKG (green), Omira et al. (2019) (red), and Widiyanto et al. (2019) (yellow), as well as the comparison index, $K$ and $\kappa$ values. $n$ stands for the total number of observations. The gray shading represents the error related to the over/underestimation, i.e., +/- standard deviation (~2.1 m).




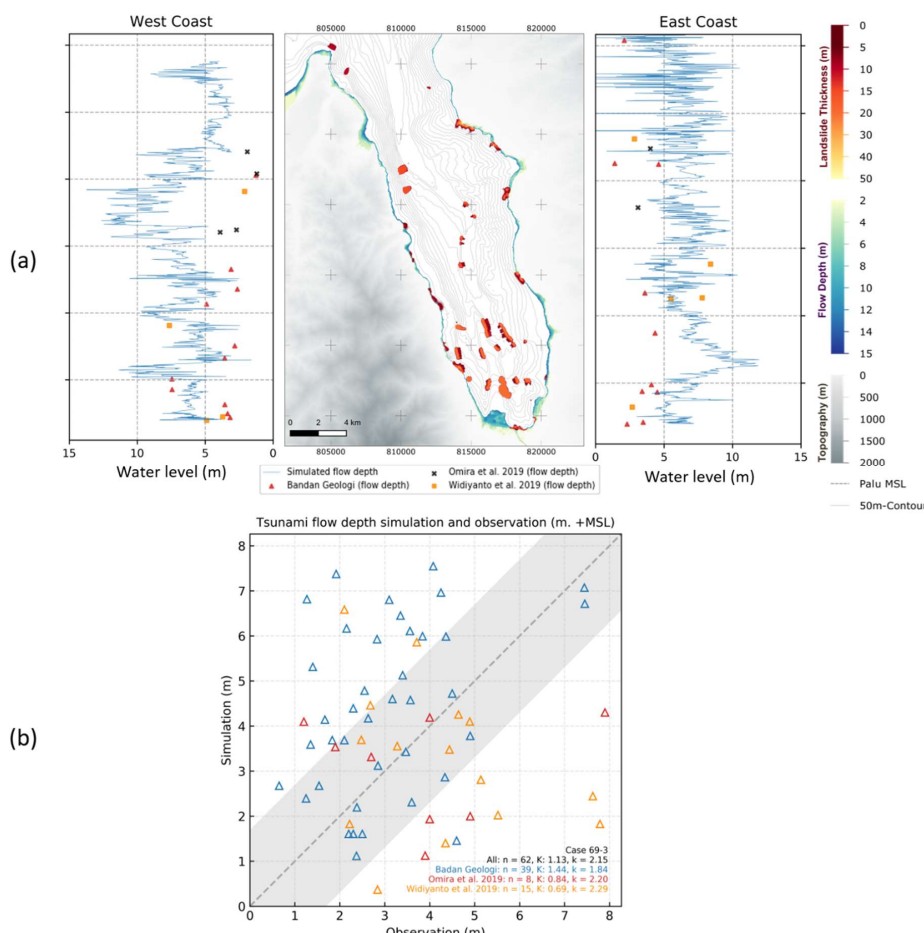

**Figure 8** Simulated tsunami inundation depth (flow depth) with the modeled submarine landslides and observed submarine landslides from previous studies. (a) Calculated maximum flow depth (center) with the maximum calculated flow depth (blue line) at the shoreline of the western coast (left) and the eastern coast (right) and the measured inundate depths from Badan Geologi's report (red triangles), Omira et al. (2019) (crosses), and Widiyanto et al. (2019) (yellow squares). (b) Comparison between simulated and observed tsunami inundation depths by Badan Geologi's report (blue), Omira et al. (2019) (red), and Widiyanto et al. (2019) (yellow), as well as the comparison index, $K$ and $\kappa$ values. $n$ stands for the total number of observations. The gray shading represents the error related to the over/underestimation, i.e., +/- standard deviation (~1.8 m).


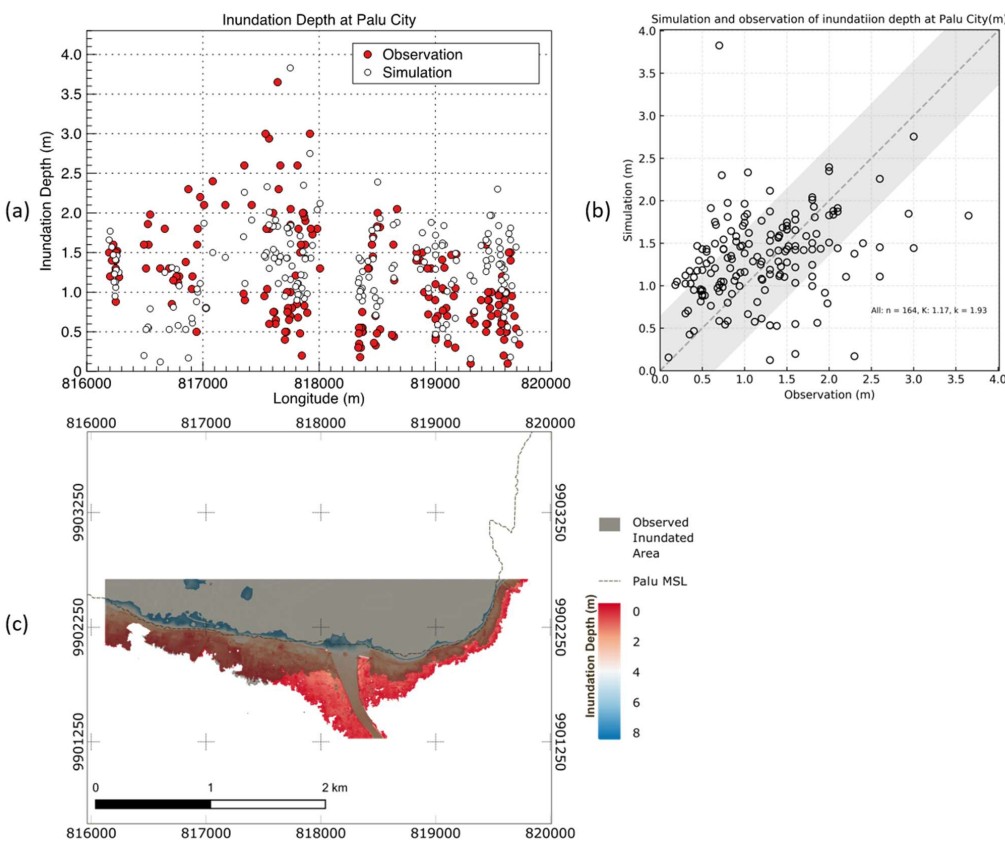

**Figure 9** Tsunami inundation depths and the area around Palu city. (a) Simulated inundation depths and observation depths by Paulik et al. (2019) plotted over longitude. (b) Comparison between the simulated and observed tsunami inundation, as well as the comparison index, $K$ and $\kappa$ values. $n$ stands for the total number of observations. (c) Simulated tsunami inundation area (in color) and the observed inundation area retrieved from Gusman et al. (2019) (in grayscale).


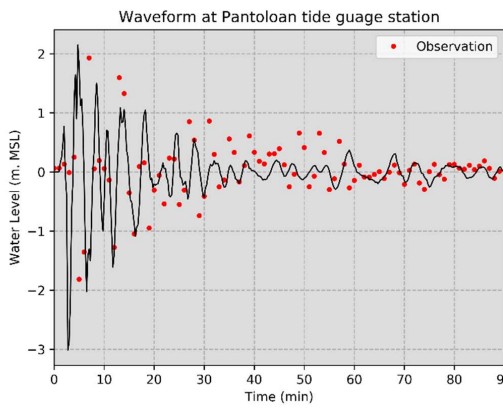






**Figure 10** Time series of sea level at the Pantoloan tide gauge with detided sea level data. The black line represents the simulated waveform, and the red dot represents the sea level record at the Pantoloan tidal gauge.

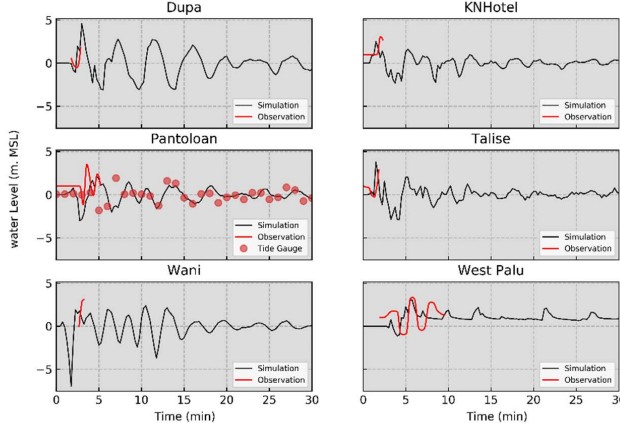

**Figure 11** Time series of sea level for the 2018 Palu tsunami at 6 locations: Dupa, the KN Hotel, Pantoloan, Talise, Wani and West Palu. The
black line represents the simulated waveform, and the red line represents the video-inferred waveform retrieved from Carvajal et al. (2019). The simulated and observed waveforms are detided and adjusted to the same model's terrain datum. The location of each CCTV is shown in Fig. 6.