# Peer review of "Submarine Landslide Source Modeling using the 3D Slope Stability Analysis Method for the 2018 Palu-Sulawesi Tsunami"

_Natural Hazards and Earth System Sciences, 2021_

## Author Comment (AC1)

**Simulation No. 12**

Date: 9/18/2020
Range of Inclination Angle: 15 – 20
Earthquake Intensity: 0.30
Surface Thickness: 6m

[Figure]

Simulated Coastal Landslides

[Figure]

Literature's Coastal Landslides

○ Matching Landslide Location ⭘ Un-Matching Landslide Location

---

## Author Response (AR1)

**Responses to Reviewer I**

| | |
|---|---|
| **Comment** | Tsunami generation by strike slip earthquakes

The USGS moment tensor of the Palu-Sulawesi earthquake is not a pure strike-slip (https://earthquake.usgs.gov/earthquakes/eventpage/us1000h3p4/moment-tensor) and thus, such a mechanism is capable of generating some tsunami waves, mainly due to the effect of the rake component. Recent studies (e.g. Elbanna et al., 2021; Frucht et al., 2019) have shown the complexity and importance of tsunamis generated by strike-slip earthquakes. Therefore, in my opinion, the potential contribution of the coseismic deformation induced by the Palu-Sulawesi earthquake to tsunami generation should not be ignored, or at least examined if relevant.

Indeed, the authors mention the need for adding coseismic sources for the modeling and the important conclusion raised by Sepúlveda et al. (2020) (lines 355-357 in the present manuscript). However, they attribute the misfit between their modeling and field observations to variations on the time of landslide initiation, etc., rather than first modeling the combined effect of tsunami generation by both coseismic deformation and subaerial and submarine landslides, and then discuss the reasons of misfit. This line of investigation is not mentioned in the Conclusions section as well. |
| **Response** | Based on the simulation by Pakoksung et al. (2019), the tsunami component calculated by Finite Fault model developed by USGS. The authors have added this tsunami components to the previous results.

Pakoksung et al. (2019) has shown the small effects of coiseismic fault to the tsunami and therefore, has unnoticeable changes or slightly changes from the past results. The tsunami waveform calculated by finite fault model were added to 10 and 11. All the results are revised by adding this tsunami induced by coseismic component. |
| **Comment** | Soil data

I found it difficult to follow the various soil layers and strata described in the text (Section 2.1.2) and Table 2 (what is the meaning of 'Underground'? Which of the base layer used for the modeling, the dry or saturated conditions?), and sketched in Figures 3 (two layers only) and Figure 5 (3 layers). |
| **Response** | In this model, soil mass was divided into 3 strata; stratum no. 1, no. 2, and the base layer as shown in Figure 5. Landslide occurs in the first 2 strata (sliding layers). The base layer does not fail. The orange sketched soil mass in Figure 3 is the same soil mass in Figure 5. Figure 3 shows the soil mass in assembly, while Figure 5 shows more details of the soil mass column.

Underground in Table 2 represents the water conditions in the model such as mean sea level, groundwater table. 'Underground' is just naming for modeling's sake. The authors have changed change to 'water level condition' in the revised version. |
| **Comment** | There is much discussion on the potential, limitations, and uncertainties of the Hovland's approach to identify the location of the submarine slope |

| | failures. Was this approach use to identify on-land and coastal slope failures that are well recognized and mapped along the Palu Bay, and thus validate the Hovland's approach for identifying potential submarine landslides |
|---|---|
| **Response** | The authors used Hovland's approach to identify the coastal landslides and validated them with observed coastal landslides as shown in Figure 1 in the manuscript. The results of our simulated coastal slope failures are presented in the attached figure. More than half of simulated landslides were found matched with the observations. However, the configurations of soil parameters in those simulations were different from measured ones as described in section 2.1.2. Coastal landslides are important for tsunami generation in this study. Thus, the authors used the coastal landslides from literature instead of our simulated coastal landslides.
[Figure]
 |
| **Comment** | General impression

The tone I felt while reading the Discussions and Conclusions sections is negative and thus discounts the achievements of this project. The critic approach displayed in these sections is highly appreciated, however it gives the impression that the project failed to achieve its goals – which in my opinion is not the case at all. I would suggest the authors to first emphasize their achievements in line with the stated goals (lines 100-108), then discus the shortcomings, and finally propose what should be done and improved next. |
| | The author has revised the discussion accordingly. |
| **Comment** | I suggest adding a general location map with an inset map that sows the study area. There is a need to add the location of the 2018 epicenter, the pattern of the seismogenic fault and surface rupture, in relation with the study area. Also, there is a need to show the various mechanisms proposed for this earthquake, because of the reasons mentioned above. |

| | Authors understand the reviewer's intention and have respect to this comment. However, the study area has already introduced in many past literatures and this study is more like the further study to them. Therefore, authors do not see the necessary to add the general location map. However, since this study added the tsunami generated by cosesimic deformation, authors have added the seismogenic fault in Fig. 6 |
|---|---|
| **Comment** | 11, 83, etc,: "visible landslides" – do you mean subaerial landslide, such that initiated on land, entered the sea and generated a tsunami? Or submarine landslides that produced visible cloudiness in the water? Please define the exact terminology in the abstract and explain it later on in the text where relevant. |
| | Authors mean observed subaerial landslides and have revised the abstract accordingly. |
| **Comment** | 16-17: "surveyed soil properties" – If I understood correctly, properties of on-land, dry soil, were extrapolated onto submarine seabed conditions with some corrections? In my intuitive understanding, the word 'soil' refers to on-land areas and 'seabed' to the upper (soil) layer in marine environment. Please define and explain your terminology, describe the process of transforming on-land dry soil properties to seabed fully saturated conditions, in short in the abstract, and later on along the text in section 2.1.2, and where else relevant: |
| | Authors rather mean the former matter. They are untouched soil measured at observation site around inland areas and eventually assumed as seabed condition. The unit weighted are slightly increase from the observation (randomly increase to get the best results).

Author have revised abstract accordingly. |
| **Comment** | 17: After describing the landslide volume, location and mechanical properties used for the modeling, one expects to see the properties used to simulate the collapse process, i.e. speed of motion, distance to rest, etc. This should also be addressed and explained in the text, especially in the methodology and Figure 2. |
| | The submarine landslides were assumed as dense fluid mass in tsunami model part and the landslides movement was describe in Pakoksung et al. (2019). |
| **Comment** | 18-19: "The results were combined with the other tsunami sources, i.e., earthquakes and observed coastal collapses,…" – I am not sure I understood correctly what exactly you mean:

Did you mean in 'results' - tsunamis induced by submarine landslide that were modeled in this study, and in 'other tsunami sources' - tsunamis simulated by other researchers due to coseismic deformation generated by the Palu-Sulawesi earthquake, as well as tsunamis induced by the observed subaerial coastal collapse? In other words, do you mean that tsunami components generated by coseismic deformation and subaerial landslides were investigated in this study?

Please clarify in the abstract and explain in details in the text. |
| | Authors mean the former description and revised the abstract and detail in section 2 Methodology. |

| Comment | 18-19: "The results were combined with the other tsunami sources, i.e., earthquakes and observed coastal collapses,…" – I am not sure I understood correctly what exactly you mean: |
|---|---|
| | Authors have changed accordingly to all specific comments where reviews marked. |
| Comment | 30: What was the tsunami type of the ninth event? |
| | The references did not mention the type of tsunami also. |
| Comment | 46-47, 50-51: Are these the reasons why tsunami component due to coseismic deformation were not simulated in this work? |
| | Yes, however, authors have added the tsunami component derived by coseismic deformation in this study and they are shown in revised Figure 6-11. According to the main comments. |
| Comment | 53: Please consider mentioning the relevant references, since this is the first time you mention the Pantoloan tide gauge record and other studies of landslide sources. |
| | Authors have revised accordingly. |
| Comment | 94: you mean previous studies of the Palu-Sulawesi event? |
| | Yes. |
| Comment | 103-104: not clear, please rephrase |
| | Authors have revised as follow.

1) *Generate the potential submarine landslide using a sophisticated landslide model based on 3D slope stability analysis (which has never been performed according to the existing literature), also based on the existing observational soil data, and to investigate whether the simulated submarine landslides match the observations or are located within potential areas suggested by past studies.* |
| Comment | 108: you mean: …with parameters calculated by tsunami simulation that are based on the developed landslide model… ? |
| | Yes, authors have revised accordingly. |
| Comment | 153: should be "… safety factor > 1" ? |
| | Yes, authors have revised accordingly. |
| Comment | 217-218: Are Upper-, Middle- and Lower- Bay refer to Northern, Central and Southern parts of the Bay? |
| | Author has changed line 172-173 for smooth reading, and the definition of upper, middle, and lower zone.

*northern, central, and southern zone (named as upper, middle and lower Palu Bay respectively, as shown in Fig. 4).* |
| Comment | 252: …in the range of ???? m error? |
| | Authors mean in the range of -2.1m to 2.1m error. |
| Comment | 267: What does it mean : "Moreover, the simulation in this study can slightly overestimate."? |
| | Authors have revised as follow.

*Moreover, the tsunami simulation results in this study can be slightly overestimated.* |

| Comment | 279-280: Figure 10 reads first apparent signal as positive wave of few cm within the first 1-2 minutes, then the first negative wave at minus ~2 m… within 5 minutes, and then the maximal positive… ? |
|---|---|
| | Author has revised accordingly as follow; *The record tsunami wave amplitude time series at the Pantoloan tidal gauge with detided sea level is depicted in Fig. 10. The first positive wave reach ~0.20 m within the first 1-2 minutes and was followed by the first negative wave at ~-1.80 m within 5 minutes. The reccord tsunami wave peak of ~1.95 m was reached at the tidal gauge within 6 minutes.* |

**Response to reviewer 2:**

| Comment | Suggest improving the quality and readability of Figures 1, 4, 5, 6, 7, and 8 |
|---|---|
| | The authors appreciate the reviewer's compliments and constructive comments. They have greatly improved and polished the content of this article. The authors have improved the quality and readability of Figures 1, 4, 5, 6, 7, and 8 in the revised version. The topography (onshore, offshore) and inundation area will be more visible and clarified in Figure 9c. |
| Comment | Define onshore and offshore in Figure 9. |
| | Author has revised the figure accordingly. |

---

## Author Response (AR2)

**Responses to Reviewer III**

| | |
|---|---|
| **Comment** | Regarding one of your references, the correct name is "Heidarzadeh". In some places in the text, figures, and captions, you have the wrong spelling for this name. |
| **Response** | The authors have revised accordingly. |
| **Comment** | Section 2.2: How have you considered landslides in your tsunami model? As static start? If yes, mention it. And, how about the timing of different landslides? Are all of them started at the same time? If yes, mention it and clarify. |
| **Response** | *How have you considered landslides in your tsunami model? As static start?* → Yes.

*how about the timing of different landslides? Are all of them started at the same time?* → Yes. They were considered starting to move at the same time. The authors have revised section 2.2 and mentioned it accordingly. |
| **Comment** | L198- 244: Section 3.1:

Here, show the locations of your landslides in a figure and name them such as slide1, slide 2,….In the current version, it is not clear how many slides you have and where they are. This is a very important issue and you need to clearly address this. Also, it would be useful to add a table for the information on the slides that you modeled. |
| **Response** | The authors have revised Figure 6 by naming our calculated landslides with no. 1 to 23, and naming landslides from the past literature with letters A to O. All the landslides in Figures 6, 7, 8 were considered in the tsunami verifications. To avoid confusion, the authors have added the explanation in section 3.1 and at the beginning of 3.2.1. |
| **Comment** | L36: I think it would be useful to mention the Anak Krakatau event as well. I suggest adding something like the following at the end of this paragraph:
"The country also experienced another tsunami in December 2018 in Anaka Krakatau killing 450 people (Muhari et al., 2019; Heidarzadeh et al., 2020)".

Heidarzadeh, M., Ishibe, T., Sandanbata, O., Muhari, A., Wijanarto, A.B. (2020). Numerical modeling of the subaerial landslide source of the 22 December 2018 Anak Krakatoa volcanic tsunami, Indonesia. Ocean Engineering, 195, https://doi.org/10.1016/j.oceaneng.2019.106733.

Muhari, A., Heidarzadeh, M., Susmoro, H., Nugroho, H.D., Kriswati, E., Supartoyo, Wijanarto, A.B., Imamura, F., Arikawa, T. (2019). The December 2018 Anak Krakatau volcano tsunami as |

| | |
|---|---|
| | inferred from post-tsunami field surveys and spectral analysis. Pure and Applied Geophysics, 176, 5219–5233. https://doi.org/10.1007/s00024-019-02358-2. |
| **Response** | The authors have added the line and references as commented. |
| **Comment** | L199: here, in order to give an overview of all landslide models, it would be useful adding something like this:

"A review of landslide tsunami models is provided by Heidarzadeh et al. (2014)".
Heidarzadeh, M., Krastel, S., & Yalciner, A. C. (2014). The State-of-the-Art Numerical Tools for Modeling Landslide Tsunamis: A Short Review. In: Submarine Mass Movements and Their Consequences, Chapter 43, 483-495, ISBN: 978-3-319-00971-1, Springer international publishing. |
| **Response** | The authors have added the line and references as commented. |
| **Comment** | Figure 7: Where are your landslides? Which ones did you consider? |
| **Response** | Where are your landslides? → our submarine landslides are ones that are far from the shore (the represented by red colors in Fig 6).

Which ones did you consider? → We considered every submarine landslide in Figure 7. Due to space limitation and avoiding the over-repetitive, figure we combined those sources of submarine landslide with the same color lamps. However, we also considered the comment by the reviewer by adding an explanation in the figure caption. |
| **Comment** | Figure 10: Connect the red dots through lines. |
| **Response** | For this comment, the author would like to keep it in 'dot' form, because the water levels at Pantoloan station are sampled at one-minute interval. The authors considered, presenting in a line might not accurately represent the observation. |
| **Comment** | Figure 11: on the figures, replace "observation" with "video-inferred". |
| **Response** | The authors have revised the figure as commented. |